# Enhancement of E-Learning Student's Performance Based on Ensemble Techniques

Abdulkream A. Alsulami [1,2,*], Abdullah S. AL-Malaise AL-Ghamdi [1,3] and Mahmoud Ragab [4,5]

1 Information Systems Department, Faculty of Computing and Information Technology, King Abdulaziz University, Jeddah 21589, Saudi Arabia; aalmalaise@kau.edu.sa
2 Department of Information Technology, Faculty of Computing and Information Technology, University of Jeddah, Jeddah 21959, Saudi Arabia
3 Information Systems Department, HECI School, Dar Alhekma University, Jeddah 22246, Saudi Arabia
4 Information Technology Department, Faculty of Computing and Information Technology, King Abdulaziz University, Jeddah 21589, Saudi Arabia; mragab@kau.edu.sa
5 Department of Mathematics, Faculty of Science, Al-Azhar University, Cairo 11884, Egypt
* Correspondence: aalsulami1207@stu.kau.edu.sa

**Abstract:** Educational institutions have dramatically increased in recent years, producing many graduates and postgraduates each year. One of the critical concerns of decision-makers is student performance. Educational data mining techniques are beneficial to explore uncovered data in data itself, creating a pattern to analyze student performance. In this study, we investigate the student E-learning data that has increased significantly in the era of COVID-19. Thus, this study aims to analyze and predict student performance using information gathered from online systems. Evaluating the student E-learning data through the data mining model proposed in this study will help the decision-makers make informed and suitable decisions for their institution. The proposed model includes three traditional data mining methods, decision tree, Naive Bays, and random forest, which are further enhanced by the use of three ensemble techniques: bagging, boosting, and voting. The results demonstrated that the proposed model improved the accuracy from 0.75 to 0.77 when we used the DT method with boosting. Furthermore, the precision and recall results both improved from 0.76 to 0.78.

**Keywords:** educational data mining; student performance; classification techniques; ensemble methods

## 1. Introduction

One of the main concerns for educational institutions is to analyze the factors that affect student performance. Every school tries to reduce the failure of their students. The most popular technique to evaluate and predict students' performance is educational data mining (EDM) [1]. EDM is about developing methods to deal with the different types of data in educational systems to improve students' learning outcomes [2] . EDM creates and modifies statistical, machine learning, and data mining approaches. EDM's primary objective is information extraction from educational data for educational decision-making. Educational data mining (EDM) can predict students' academic achievement early [3]. Their use could enhance the analysis of students' learning processes while taking into account how they interact with the environment. In this study, we investigated the electronic learning (E-learning) data set. E-learning is a field that has dramatically increased recently. Organizations and teachers have identified some challenges in E-learning. One of the main ones is to identify the factors that affect student performance while taking online courses. Therefore, we used a data set with features that allow us to analyze such factors and predict most that affect the performance. The data set has different characteristics such as demographic features, academic background, and behavioral features during taking online classes. Then, a proposed model was applied to the data set to analyze and

predict student performance. The model used three traditional data mining techniques to produce a performance model. The techniques are decision trees, naive Bayes, and random forests. Then, two ensemble methods have been used to enhance the results of traditional data mining techniques: bagging and boosting. Furthermore, each ensemble method has included two and three classifications mentioned above, respectively, with the help of voting processes for a more accurate prediction.

## 2. Related Work

This study aims to provide a comprehensive model for investigating e-learning students' data, specifically to help educational organizations in forecasting student success. There have been studies examining student performance prediction using previous educational results to predict future results at the same level. Moreover, other studies have examined the factors that affect student performance. The following is a review of some of these studies. In [4], researchers aimed to predict the student's success on the exam. They modeled the study by using the decision tree and K-nearest neighbor. The study concluded that the decision tree predicts students' pass or fail status in an academic course with the best results. In [5], researchers compared different classification techniques: Naive-BayesSimple, multilayer perceptron, SMO, J48, and REP-tree—in the field of comparing student performance. The data set was collected from a computer science department of a college with 300 student records. WEKA was the tool to use in the study. From the results, researchers concluded that the performance of multilayer perceptron is the most effective algorithm for predicting student performance. In comparison to other algorithms, multilayer perceptron accuracy was higher than other classifiers. In [6], researchers aimed to determine the basic factors that have a significant effect on secondary student performance. To do so, they combined single and ensemble-based classifiers to create the proper classification model, which they then used to forecast academic success. In the beginning, three data mining methods were used: decision tree, multilayer perceptron (MLP), and a PART; moreover, three ensemble techniques (multi-boost, bagging (BAG), and voting) were used individually. To improve the previous classifiers' performance, a single classifier and an ensemble classifier were combined, generating nine new models. According to the evaluation's findings, multi-boost with MLP outperformed other approaches in terms of accuracy. In [7], using data mining techniques, researchers were trying to predict student dropout. The findings demonstrated the possibility of dropout with accuracy rates greater than 0.80 in most situations and false positive values varying from 0.10 to 0.15 on average. K-nearest neighbors, random forest, support vector machines, decision trees, logistic regression, and naive Bayes were among the methods they were contrasting. Random forest outperformed other machine learning techniques in terms of accuracy, F-measure, and precision. In [8], researchers have concentrated on forecasting student performance in several interactive online sessions by exploring the information gathered using the E-learning and design suite. The data set keeps track of student participation during classes, including text editing, keystrokes, and the amount of time spent on each assignment. They used five well-known classifiers: naive Bayes, random forest, support vector machine, multi-layer perception, and logistic regression. Three distinct evaluation methods were utilized: five-fold cross-validation and random data split for training and testing. The model was trained in all sessions except the one used for testing. According to the results, the RF classifier model obtained the best accuracy. In [9], researchers investigated various classifier algorithms that are proposed to predict secondary school students' success in mathematics and Portuguese lessons. They classified using support vector machine (SVM), linear discriminant analysis (LDA), and K-nearest neighbor (KNN). Their experimental results demonstrated that the SVM method performed better for the unbalanced class distribution problem. In [10], the classification technique being evaluated by researchers was a hybrid classification. To do so, they used the radial basis function network, C4.5, random forest, and multilayer perceptron algorithms. They observed that hybrid classification algorithms perform more accurately than single algorithms. In [11],

researchers were clustering the data by using the K-nearest neighbors (KNN) algorithm with the help of Harris hawks optimization (HHO). Once they classify all of the solutions, redistribution for the solutions into a search space will be applied. Several different machine learning classifiers were used to validate the overall prediction system, such as naive Bayes, KNN, LRNN, and artificial neural network. The results collected demonstrate the significance of anticipating student performance early to reduce student failure and enhance the overall effectiveness of the educational institution. Furthermore, given that LRNN is a deep learning method that can observe past and current input values, the results showed that the modified HHO and LRNN combination outperforms other classifiers with an accuracy of 0.92. In [12]. Researchers concentrated on how crucial it was to take advantage of both technological advancements and potential educational contributions. They tested a new PFA strategy based on various ensemble learning techniques to improve the forecasting of student performance. (random forest, AdaBoost, and XGBoost). The results have demonstrated that XGBoost could predict future student acquisition with the highest performance. In [13], researchers presented the data mining technique used to forecast first-year students' academic performance. They chose three different data models for learning stages and tested them based on the dates of entry, end of the first, and end of the last semesters. Records of bachelor students who enrolled in a program offered by the institution between 2006/2007 and 2015/2016 were obtained and gathered through the institutional database. The best overall performance was gained by a support vector machines (SVM) model, which was chosen to perform database sensitivity analysis. Table 1 shows some papers that used different data mining techniques in order to predict the performance of students.

**Table 1.** Comparison of data mining techniques in predicting student's performance.

| Year, Author(s) | Methodology | Key Findings |
| --- | --- | --- |
| 2022, Aremu, Dayo Reuben, Awotunde, and Ogbuji [4] | Decision tree (DT) and K-nearest neighbor (KNN) | Decision tree DT for predicting the pass/fail status of students delivers the most successful outcomese |
| 2021, Siddique, Ansar, et al. [6] | MLP, J48, and PART BAG, MB, and VT | Concluded MultiBoost with MLP outperformed the others. |
| 2021, Palacios, Carlos A., et al. [7] | DT, KNN, LR, NB, RF, and SVM | RF algorithm ranked first among the others. |
| 2022, Begum, Safira, and Sunita S Padmannavar [8] | KNN, LDA, and SVM | Shown that the SVM is for the unbalanced class distribution problem. |
| 2022, Brahim, Ghassen Ben [9] | MLP, RF, SVM, NB, and LR | Showed the best classification accuracy performance |
| 2021, Kumar, A. Dinesh, R. Pandi Selvam, and V. Palanisamy [10] | multilayer perceptron, Radial basis function network, C4.5, and random forest algorithm | Hybrid classification algorithms are more accurate than individual classification algorithms. |
| 2021, Gil, Paulo Diniz, et al. [13] | DT, RF, SVM, and ANN | SVM has the highest accuracy among others |
| 2022, Joshi, Manuj, and Chawda [14] | NaiveBayesSimple, Multilayer Perception, SMO, J48, and REPTree | Concluded multilayer perception algorithm is most appropriate for predicting student performance. |
| 2021, Ahammad, Khalil, et al. [15] | support vector machine, naive Bayes, K-nearest neighbours, XG-boost, and multi-layer perceptron | Multi-layer perceptron achieved the highest accuracy |

In [16], researchers attempted to determine the factors influencing academic performance. Thus, they made use of two different types of data sets. The first data set demonstrates how the performance in a course's required courses might affect a student's performance in the current course. The second data set suggested that the student's grade in any course is related to their performance in the semester until the midterm test. In [17], the results of the model showed that the main contribution to predicting academic performance is related to the following factors: interview, task, questionnaire, and age. The access factor measures student's access to the module, including access to forums and

glossaries. Questionnaire factors summarize the variables in the questionnaire related to the visit and the attempt. The age factor contains the student's age. In [18], the study aimed to investigate the factors affecting student performance. Researchers reviewed and analyzed 36 articles. They concluded that the performance in previous classes and grades, the students' e-Learning activities, and their demographic background had an impact on the performance of the student, academically speaking. In order to determine whether students' learning behaviors were important, researchers examined the same data set [19]. They used the ensemble methods, voting, bagging, and boosting, alongside traditional data mining methods, support vector machines, decision tree (ID3), K-nearest neighbor, and naive Bayes. With the help of the voting process, the highest accuracy was achieved. In [14], an investigation was conducted on learners' relationships with e-learning. A combination of ensemble algorithms with three different types of classifiers was used: decision trees, K-nearest neighbors, and support vector machines. It was found that learners' features were strongly correlated with their performance in the study. In contrast, ensemble techniques increased accuracy. In [15], in order to help decision-makers make the best choices for their organizations, researchers used ensemble methods to predict student performance. They used naïve Bayes, decision tree, and K-nearest neighbor methods. The voting technique was used to combine the three methods. In most scenarios, the proposed model improved the accuracy of naïve Bayes.

## 3. Methodology

In this section, we describe the data set used to conduct this study, followed by a discussion of the proposed model and the evaluation measures.

### 3.1. Data Set

#### 3.1.1. Data Collection

The data set for this study was obtained from the Kalboard 360 E-Learning system via the Experience API (XAPI). The data set [20] in this study consists of 480 records with 17 attributes. In addition, all attributes are either integer or categorical in nature. The features are categorized into three major types. Table 2 shows (1) demographic attributes such as place of birth, nationality, gender, and parent responsibility for their children. (2) Academic attributes such as educational stage and grade level. (3) Behavioral attributes include opening resources and raising hands in class. These different categories make the dataset appropriate for the classification and prediction of student performance within E-learning systems. Table 2 illustrates the wide range of data set features, along with the category to which they belong, in addition to the description of each attribute.

**Table 2.** Classification results with traditional DM.

| Evaluation Measures | DT | NB | RF |
|:---:|:---:|:---:|:---:|
| Accuracy | 75.5% | 67.7% | 76.6% |
| Precision | 0.760 | 0.675 | 0.766 |
| Recall | 0.758 | 0.677 | 0.767 |
| F-Measure | 0.759 | 0.671 | 0.766 |

#### 3.1.2. Data Visualization

Data visualization is an essential part of the preprocessing process that uses graphs to simplify complex data. We used WEKA software to visualize the data set. The graphical representations can help instructors better understand their students and monitor what's happening in online classes. Figure 1 illustrates the gender. The data set consists of 305 males and 175 females.

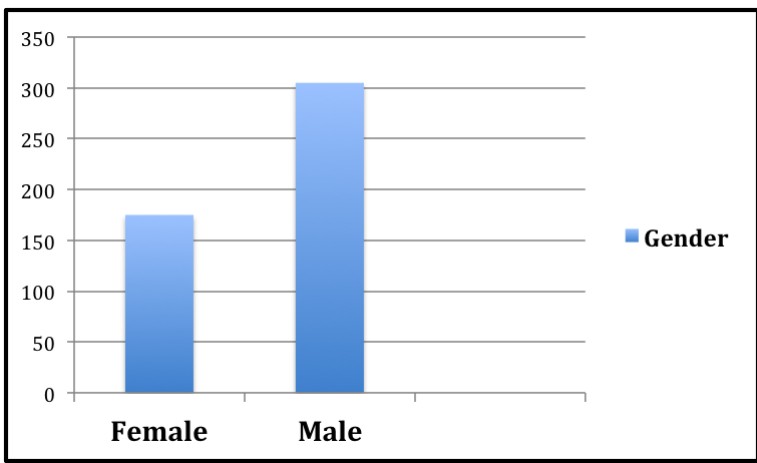

**Figure 1.** Student gender.

Figure 2 shows the diversity of nations that the data set contains. For example, 179 students are from Kuwait and 172 students are from Jordan and others.

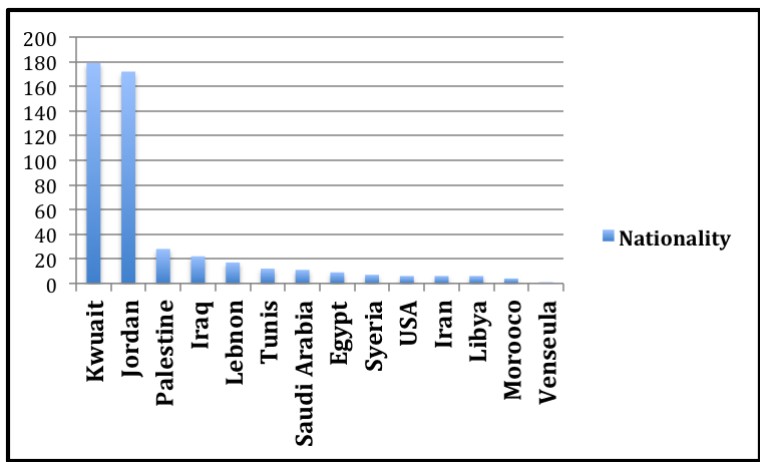

**Figure 2.** Student nationalities.

The data set also includes a feature for recording students' attendance at school to measure the influence of such a feature. As seen in Figure 3, students are divided into two groups according to the number of absence days: 289 students have fewer absence days than 7, while 191 students have more than 7.

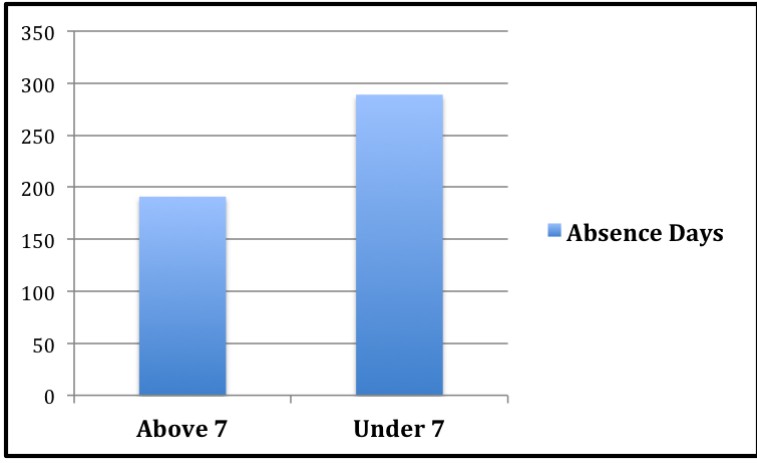

**Figure 3.** Student absences.

Students are classified into three educational levels based on the characteristics listed: there were 199 lower-level students, 248 middle-level students, and 33 high-level students. Students' records were gathered over the courses of two academic semesters: the first and second, with 245 records being obtained from the first semester and 235 records collected from the second. Students chose various topics across these semesters. In total, 95 students were taking the IT topic, 65 were taking the French topic, 59 were taking the Arabic topic, 59 were taking the Arabic language, 51 were taking science, 45 were taking English, 30 were taking biology, 25 were taking Spanish, 24 were taking both chemistry and geology, 22 were studying the Quran, 21 were taking math, and 19 were taking history.

### 3.1.3. Data Cleaning

As part of preprocessing, data cleaning is essential for removing irrelevant objects and missing values in the data collection. There are zero missing values in the data set.

### *3.2. Features Selection*

Feature selection refers to selecting the relevant features of a dataset based on specific criteria from an original feature set. There are two types of data reduction methods: wrapper methods and filter methods. The filter method ranks the features using variable ranking methods, with the highly ranked features being selected and implemented into the learning algorithm [21]. In this study, the information gain ranking filter and a correlation-ranking filter were used. At each decision tree node, and in order to select the test attribute, the information gain measure is taken into account. The information gain (IG) metric determines features with a large number of values. It is calculated with Equation (1).

$$IG(T, a) = H(T) - H(\mathrm{T}|a) \tag{1}$$

where $T$ is a random variable and $H(T|a)$ is the entropy of $T$ given the value of attribute a.

Figure 4 illustrates the feature ranking after using the WEKA tool to apply the information gain filter. Visited resources are ranked first, followed by student absence, raised hands, and other attributes.

```
        Information Gain Ranking Filter

Ranked attributes:
 0.45801  11 VisITedResources
 0.39745  16 StudentAbsenceDays
 0.37337  10 raisedhands
 0.2578   12 AnnouncementsView
 0.1504   14 ParentAnsweringSurvey
 0.12773   2 NationalITy
 0.1261    9 Relation
 0.12292   3 PlaceofBirth
 0.11393  13 Discussion
 0.10676  15 ParentschoolSatisfaction
 0.07611   7 Topic
 0.05178   1 gender
 0.04748   5 GradeID
 0.01182   8 Semester
 0.01058   4 StageID
 0.00703   6 SectionID

Selected attributes: 11,16,10,12,14,2,9,3,13,15,7,1,5,8,4,6 : 16
```

**Figure 4.** Information gain filter.

Correlation coefficients are applied to measure correlations among attributes and classes and inter-correlations between features [21]. It is calculated with Equation (2).

$$\rho(X, Y) = \frac{\text{cov}(X, Y)}{\sigma_X \sigma_Y} = \frac{\sum_{i=1}^{n}(x_i - \bar{x})(y_i - \bar{y})}{\sqrt{\sum_{i=1}^{n}(x_i - \bar{x})^2}\sqrt{\sum_{i=1}^{n}(y_i - \bar{y})^2}} \tag{2}$$

where:
- $X$ and $Y$ are the two variables being correlated;
- $n$ is the number of data points;
- $x_i$ and $y_i$ are the values of $X$ and $Y$ for the data point;
- $\bar{x}$ and $\bar{y}$ are the means of $X$ and $Y$;
- $\text{cov}(X, Y)$ is the covariance between $X$ and $Y$.

Figure 5 shows the rank between the attributes and the class. As we saw in the information gain filter, the most ranked feature is the visited resources feature, which is followed by student absence days and raised hand. From Figures 4 and 5, the filters selected 16 attributes. As a result, the behavior feature impacts student performance more than the other features due to its significant impact.

```
         Correlation Ranking Filter
Ranked attributes:
 0.3829  11 VisITedResources
 0.3608  16 StudentAbsenceDays
 0.3283  10 raisedhands
 0.2895  12 AnnouncementsView
 0.2369  14 ParentAnsweringSurvey
 0.2358   9 Relation
 0.1852  15 ParentschoolSatisfaction
 0.1508  13 Discussion
 0.1266   1 gender
 0.088    3 PlaceofBirth
 0.078    2 NationalITy
 0.0673   4 StageID
 0.065    8 Semester
 0.0487   7 Topic
 0.0459   5 GradeID
 0.0344   6 SectionID

Selected attributes: 11,16,10,12,14,9,15,13,1,3,2,4,8,7,5,6 : 16
```

**Figure 5.** Correlation filter.

### 3.3. Data Mining Tool

WEKA is a well-known Java-based machine learning software developed at New Zealand's University of Waikato [22]. The WEKA package includes visualization tools, data analysis and predictive modeling algorithms, and graphical user interfaces for better accessibility to this functionality. It includes numerous algorithms for data mining and machine learning.

### 3.4. Proposed Model

The primary purpose of this paper is to compare the performance and results of each prediction model based on the use of traditional data mining techniques and ensemble methods. Figure 6 illustrates the proposed model that will apply to the data set. First, we collect the data set and prepare it to perform the study. Then, three traditional data mining methods will apply (decision tree (DT), naïve Bayes (NB), and random forest (RF)) to produce a performance model. In addition to the classifiers mentioned earlier, two ensemble methods are used to improve their performance. Boosting, as well as bagging, is applied to enhance the student prediction model's success. Two and three methods were

added to each ensemble technique using the voting process for a more accurate prediction. The model's last phase will involve evaluating and discussing the results. The data were divided into training and test sets. Each prediction model's performance was evaluated using K-fold cross-validation. When testing a model, this technique is used to solve the variance problem. In brief, k-fold cross-validation divides the training set into 10 folds. During training, 9 folds are applied before the final fold is tested. As an average of the different accuracies is taken, this better represents the model performance. The method was repeated ten times. All models were run with the WEKA software's default parameters.

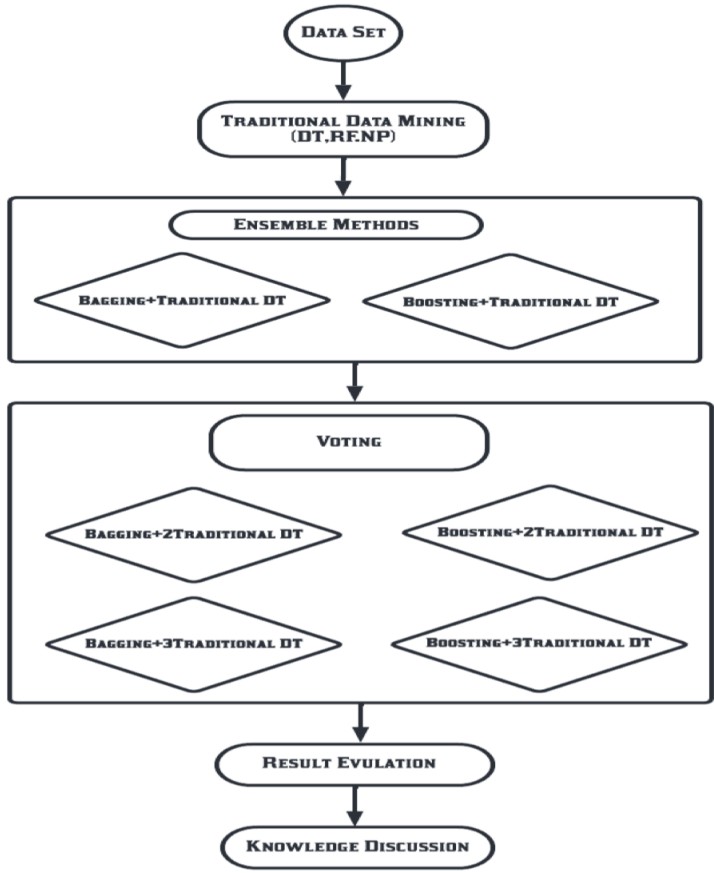

**Figure 6.** Proposed model for predication of student's performance.

*3.5. Description of Traditional Data Mining*

3.5.1. Decision Tree

The decision tree algorithm belongs to the family of supervised learning algorithms. Decision trees are one of the most effective techniques in various fields, including machine learning, image processing, and pattern recognition. The decision tree algorithm solves both regression and classification problems. By learning basic decision rules from training data, researchers can create training models that predict the class of a specific variable. Each tree is composed of nodes and branches. A feature in the classification category identifies each node, and each subset identifies a value the node can take [23].

3.5.2. Random Forest

In machine learning, random forest is a supervised algorithm that predicts output by merging multiple decision trees into a forest, and then by combining the predictions from all decision trees using ensemble learning, an accurate prediction can be obtained [23].

### 3.5.3. Naive Bayes

According to Bayes' theorem from probability theory, naive Bayes is a direct and simple Bayesian classifier. An NB algorithm applies Bayes' theorem to each pair of features given the class variable's value and assumes that they are conditionally independent [24]. Every pair of features being classified in naive Bayes is independent of the others.

### *3.6. Description of Ensemble Methods*

### 3.6.1. Bagging

Bagging is the most well-known independent method. The method aims to improve accuracy by combining the results of multiple learned classifiers into a single prediction to create an improved composite classifier. The classifiers are trained with replacements on subsets of instances from the training set [25]. There is no difference in sample size between the original training set and each sample. Bagging is a technique for improving a classifier's accuracy by creating multiple classifiers and combining multiple models. The bagging method aims to improve the accuracy of unsteady classification models by constructing a composite classifier and then combining the results of the obtained classification models into a single prediction. It means that every set of data has an equal chance of being taken [26].

### 3.6.2. Boosting

Boosting refers to a group of algorithms that can modify weak learners into strong ones. Boosting works by training multiple classifiers and obtaining their forecasts, then adjusting the weights of the weakest one to reduce the previous learner's errors. Boosting was used only for binary classification. The AdaBoost algorithm overcomes this adaptive limitation. Boosting determines instances according to their weight with a possible [27].

### 3.6.3. Voting

A voting classifier is a machine learning model where the class is predicted based on the output with the greatest probability based on different base models. The voting procedure involves learning classifier voting by a majority (for classification) and average (for regression). Eventually, the highest vote or average obtained for each class will be predicted [28].

An independent process or a dependent process can be classified as a set mode. A dependent method is considered to be boosting. Learners are created based on their outcomes in a dependent process. Each learner works independently during the independent process, and their outcomes are merged using a voting process. The bagging method is an independent method [29].

### *3.7. Measurement Measures*

In this study, data were applied to the WEKA Data Mining tool. Data were fed into the WEKA Data Mining tool in this study. Then, different DM techniques were compared to determine which had higher prediction accuracy than others, and a decision was made based on that. The following common metrics can evaluate a study's performance: accuracy, precision, recall, and F-Measure.

### 3.7.1. Accuracy

This represents the classifier's accuracy and relates to the classifier's capacity. The accuracy of a predictor relates to the way it accurately predicts the impact of a predicted feature for new information. The percentage of correct predictions divided by the total number of predictions yields the accuracy [30]. It is calculated with the following Equation (3):

$$Accuracy = (TP + TN)/(TP + TN + FP + FN), \tag{3}$$

where:

- True positives (*TP*): cases that are predicted as yes.
- True negatives *(TN)*: cases that are predicted as no.
- False positives (*FP*): cases that are predicted yes and are actually yes.
- False negatives (*FN*): cases that are predicted as no but are actually yes.

### 3.7.2. Precision

Precision is calculated as the ratio of correctly classified positive predictions to total positive predictions, whether correctly or incorrectly classified [30]. It is calculated with Equation (4).

$$Precision = TP/(TP + FP), \tag{4}$$

### 3.7.3. Recall

The recall is determined by calculating the proportion of correctly classified positive predictions to all positive predictions [30]. It is calculated with Equation (5).

$$Recall = TP/(TP + FN), \tag{5}$$

### 3.7.4. F-Measure

F-measure conveys both recall and precision in a single measure [30]. It is calculated with Equation (6).

$$F1 - measure = (2 * Recall * Precision)/(Recall + Precision), \tag{6}$$

## 4. Experimental Results

The results of each of the prediction models (traditional data mining without and within ensemble methods) are provided in this section.

### 4.1. Traditional DM Techniques

WEKA software was used to conduct the experiments, as we mentioned in the previous section. In Figure 7, implantation for the traditional DM model is explained. First, in WEKA, a data set was uploaded using an operator called CSVLoader to start building the model. The description and details of the selected dataset were explained in the previous section. The dataset contained 17 attributes in total, and all of them were chosen for this study. By linking the "CSVLoader" to a text viewer in WEKA, a table of all the attributes can be shown. Secondly, the data set was assigned to an operator called "ClassAssigner "to determine the attribute to be considered the class. Third, once the class had been defined, the dataset was connected to the cross-validation operator called in WEKA, "CrossValidationFoldMaker". It was divided into two parts: training and testing. In each iteration of the cross-validation process, nine subsets were trained for the model and one for the test. As a result, the model's training and validation were performed concurrently in a single step, which was recognized as a valid test as a result the dataset being used for testing is unidentified. After that, as Figure 7 illustrates, the cross-validation was connected with each algorithm. The data set was divided into two parts: training, to train the algorithm with the training data, and testing, to test the algorithm with the testing data. Finally, the "Classifier PerformanceEvaluator" was applied to each to get the validation of the model and the performance. Moreover, all the models in Figures 8 and 9 were performed with the same procedure as in Figure 7.

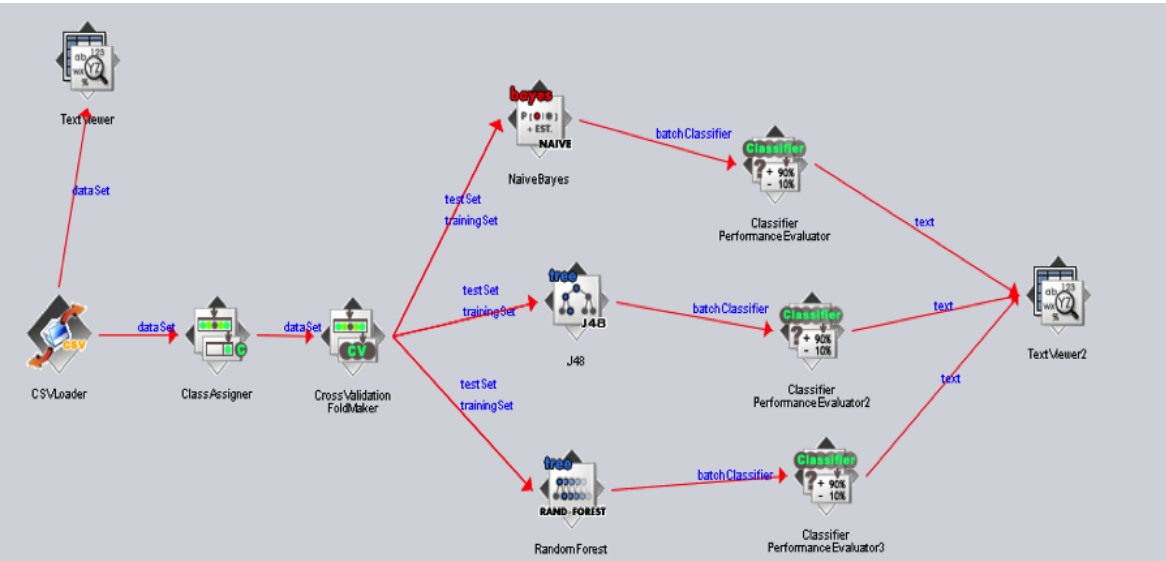

**Figure 7.** Traditional DM technique implementation using WEKA.

### 4.2. Ensemble Methods

The same procedure will be performed on the boosting model. Uploading the data set then assigning it to a class and connecting it to the cross-validation to link it to the methods and finally applying the model. As Figure 8 shows, six experiments have been performed in the boosting model. First: three experiments with each of the data mining techniques. Second, with the help of voting methods, boosting was performed with two algorithms at one time. There were two experiments: boosting with naïve Bayes and random forest, and the second was boosting with random forest and decision trees. Third, with the help of the voting method, the last experiment was conducted using boosting with all three traditional data mining techniques. The idea was to observe any difference in the performance when we manipulate the model.

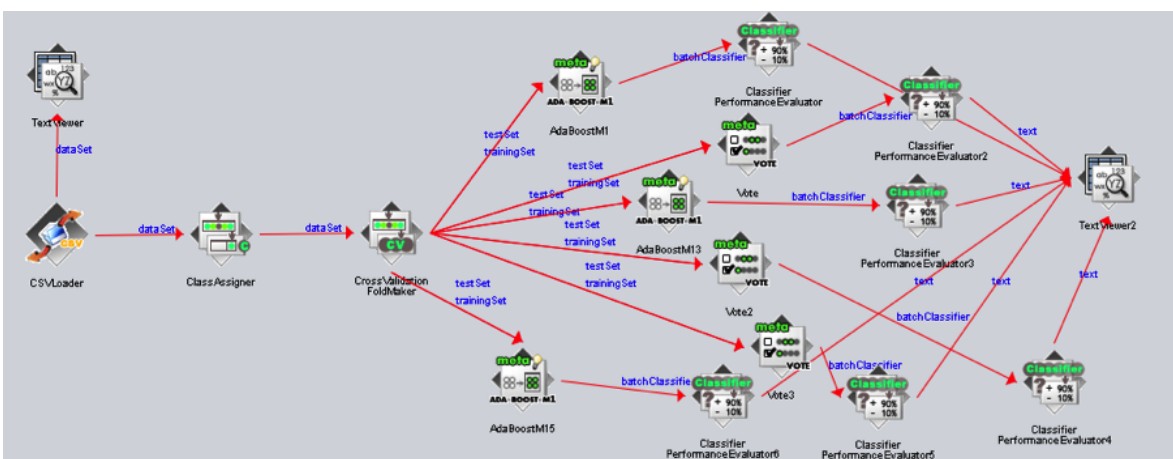

**Figure 8.** Ensemble method (boosting) implementation using WEKA.

The same procedure as discussed in boosting implementation will be used with the bagging model as seen in Figure 9.

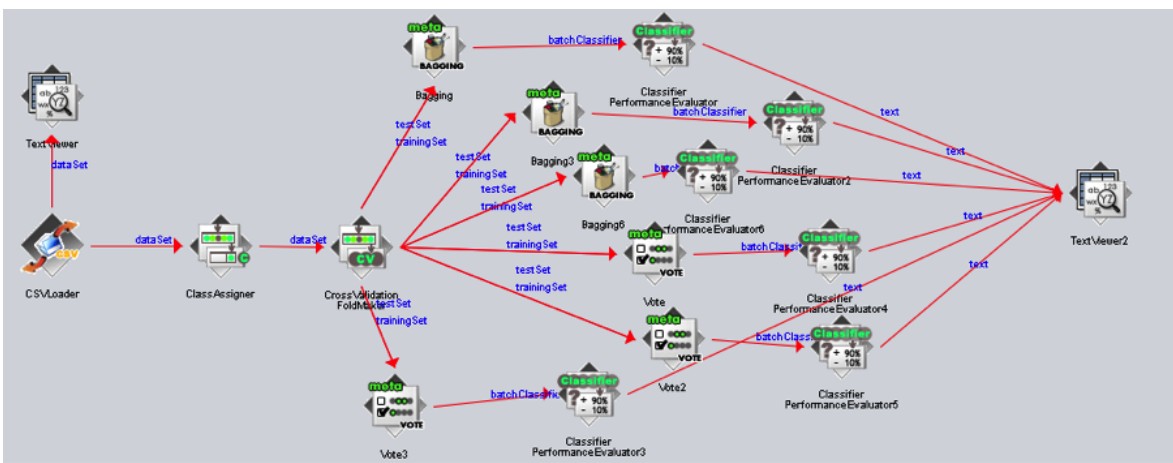

**Figure 9.** Ensemble method (bagging) implementation using WEKA.

Table 2 illustrates the evaluation measures for traditional data mining. The decision tree resulted in an accuracy of 75.5%, a precision of 0.760, a recall of 0.758, and an F-measure of 0.759. The accuracy was 67.7%, the precision was 0.675, the recall was 0.677, and the F-measure was 0.671, when using naïve Bayes. The accuracy of random forest was 76.6%, the precision was 0.766, the recall was 0.766, and the F-measure was 0.766.

Table 3 illustrates the evaluation measures for boosting with the traditional data mining techniques. Using boosting with decision trees resulted in accuracy of 77.92%, precision of 0.779, recall of 0.7779, and F-measure of 0.779. The results of using boosting with random forest were 76.25% accuracy, 0.763 precision, 0.763 recall, and 0.762 F-measure. The accuracy of 72.29%, precision of 0.724, recall of 0.723, and F-measure of 0.723 are the results of using boosting with naïve Bayes. Boosting with naïve Bayes and decision trees resulted in an accuracy of 76.45%, precision of 0.7264, recall of 0.765, and F-measure of 0.764. Finally, when using boosting with all traditional data mining techniques, the accuracy was 76.25%, the precision was 0.762, the recall was 0.763, and the F-measure was 0.762.

**Table 3.** Classification results with boosting.

| Evaluation Measures | Boosting | | | | | |
|---|---|---|---|---|---|---|
| | **DT** | **RF** | **NB** | **DT + NB** | **RF + DT** | **NB + RF + DT** |
| Accuracy | 77.92% | 76.25% | 72.29% | 76.45% | 77.29% | 76.25% |
| Precision | 0.779 | 0.763 | 0.724 | 0.764 | 0.775 | 0.762 |
| Recall | 0.779 | 0.763 | 0.723 | 0.765 | 0.773 | 0.763 |
| F-measure | 0.779 | 0.762 | 0.723 | 0.764 | 0.733 | 0.762 |

Table 4 illustrates the evaluation measures for bagging with traditional data mining. The results of using bagging with decision trees are 74.37% accuracy, 0.744 precision, 0.743 recall, and 0.743 F-measure. The results of using bagging with random forest are 75.63% accuracy, 0.757 precision, 0.756 recall, and 0.756 F-measure. Bagging with naïve Bayes produces accuracy of 67.7%, precision of 0.677, recall of 0.676, and F-measure of 0.672. Bagging with naive Bayes and decision trees achieves accuracy of 75.62%, precision of 0.756, recall of 0.756, and F-measure of 0.756. The results of using bagging with random forest are 76.46% accuracy, 0.766 precision, 0.765 recall, and 0.765 F-measure. Finally, when using bagging with all three data mining techniques, the accuracy is 76.87%, the precision is 0.768, the recall is 0.769, and the F-measure is 0.768.

**Table 4.** Classification results with bagging.

| Evaluation Measures | Bagging | | | | | |
|---|---|---|---|---|---|---|
| | **DT** | **RF** | **NB** | **DT + NB** | **RF + DT** | **NB + RF + DT** |
| Accuracy | 74.37% | 75.63% | 67.7% | 75.62% | 76.46% | 76.875% |
| Precision | 0.744 | 0.757 | 0.677 | 0.756 | 0.766 | 0.768 |
| Recall | 0.743 | 0.756 | 0.676 | 0.756 | 0.765 | 0.769 |
| F-measure | 0.743 | 0.756 | 0.672 | 0.756 | 0.765 | 0.768 |

## 5. Evaluation Results and Finding

In this section, the results of different traditional data mining techniques (decision tree, naïve Bayes, and random forest) and ensemble methods (bagging, boosting, and voting) will be interpreted and evaluated. As mentioned above, four different measures will be used to evaluate the performance: accuracy, precision, recall, and F-measure.

### 5.1. Accuracy

For all of the experiments that we conducted with traditional data mining techniques and ensemble methods, the accuracy values were above 65%. We observed that naïve Bayes had the lowest accuracy of 67.7% among all methods. On the traditional DM techniques, as shown in Figure 10, the random forest has a more notable high accuracy of 76.6% than other techniques, which indicates that 365 of 480 students were successfully classified according to the suitable class labels and 115 were not.

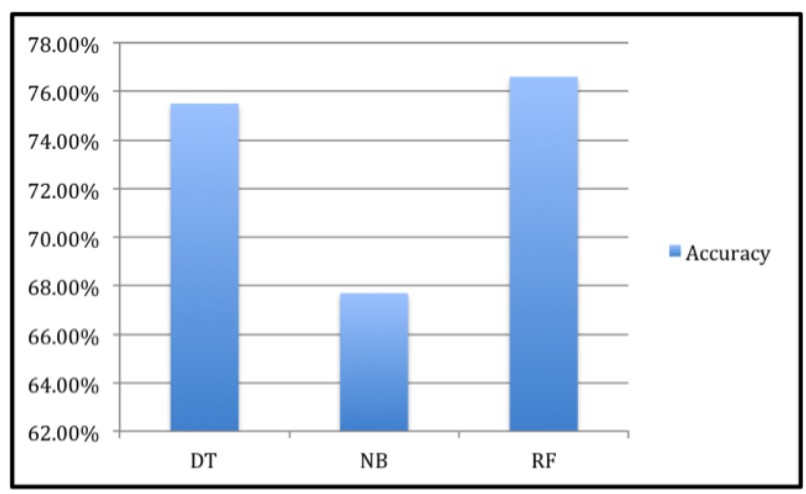

**Figure 10.** The accuracy for traditional DM techniques.

Moreover, Tables 3 and 4 explained all of the accuracy performances for the models using ensemble methods. Overall, the accuracy of performance was improved by using ensemble methods. In naïve Bayes, the accuracy was enhanced from 67.7% (without ensemble) to 72.29% (with boosting). In comparison between bagging and boosting using voting processes to traditional DM techniques, the accuracy of boosting with decision tree achieved the highest value of 77.9%, where the value was 75.5% without ensemble methods, as Figure 11 shows. That is, from 360 to 375 students were successfully classified to the appropriate class labels. Every ensemble method scenario has outperformed the naïve Bayes accuracy except with boosting, which was equal. Figure 11 illustrates how the proposed model enhanced the accuracy effectively when we used ensemble methods with the traditional data mining separately (boosting + DT) and when we combined several classifiers (Boosting + DT + RF) using the voting process.

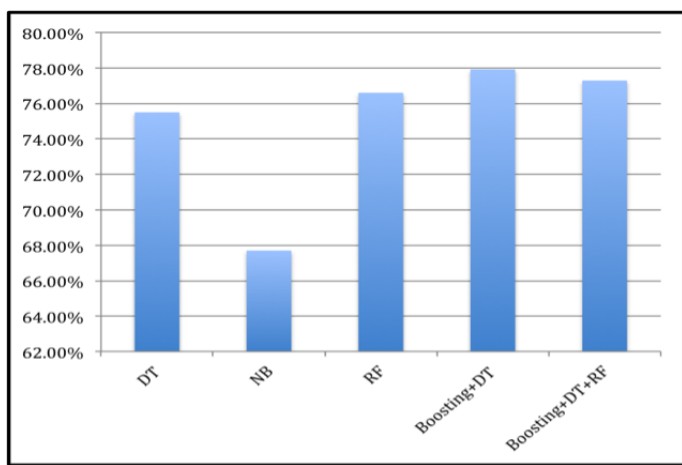

**Figure 11.** The accuracy of DM techniques and ensemble methods.

*5.2. Precision*

All fifteen experiments have a value of precision above 0.65. Figure 12 shows the precision of traditional DM techniques. We can observe that random forest outperformed the other methods with a value of 0.75.

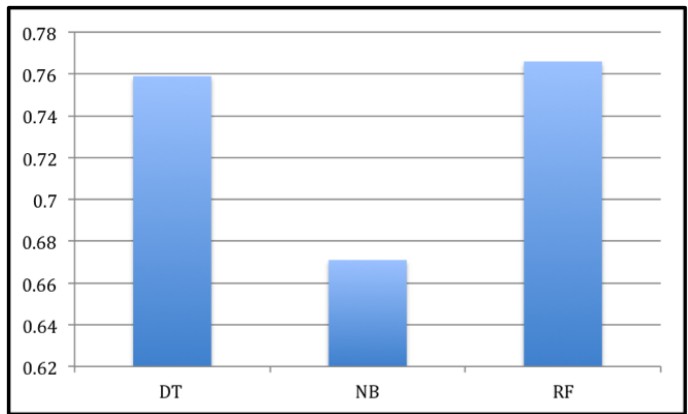

**Figure 12.** Precision for DM techniques.

Naïve Bayes has the lowest value of 0.67. At the same time, it increased when applying ensemble methods to 0.72 (with boosting), meaning the number of students correctly classified to the right class labels improved from 322 to 346. Boosting with a decision tree recorded the highest value of precision with 0.78 when in traditional data mining, and the highest value was 0.76. Furthermore, other ensemble classifiers outperformed the traditional DM techniques, as shown in Figure 13.

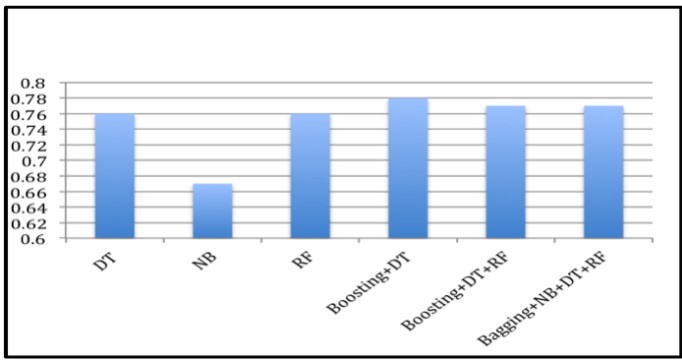

**Figure 13.** The precision of DM and ensemble methods.

*5.3. Recall*

All fifteen experiments have a value of precision above 0.65. Compared to traditional DM techniques, random forest performed better in recall than others with a value of 0.76, as shown in Figure 14.

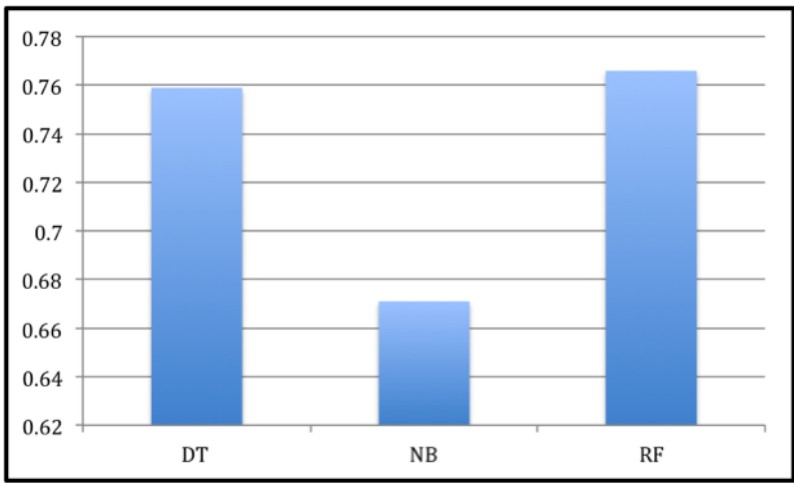

**Figure 14.** Recall for DM techniques.

Naïve Bayes has the lowest value of 0.67, while it increased, when ensemble methods were applied, to 0.72 (with boosting), which is the percentage of correctly classified students to the total number of unclassified, and correctly classified classes improved from 322 to 346. Boosting with a decision tree, as shown in Figure 15, achieved the highest value of recall with 0.78 compared to 0.76 in traditional DM.

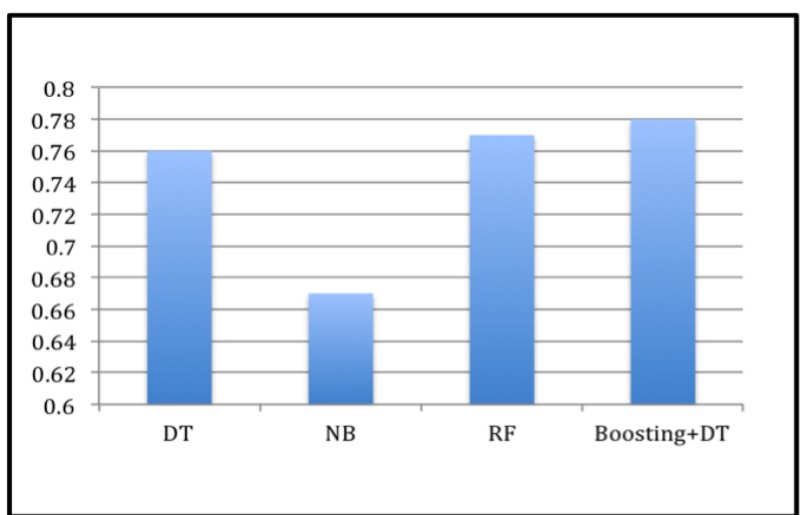

**Figure 15.** The Recall of DM techniques and ensemble methods.

*5.4. F-Measure*

All fifteen experiments have a value of precision above 0.65. Random forests and decision trees have similar values, as Figure 16 illustrates.

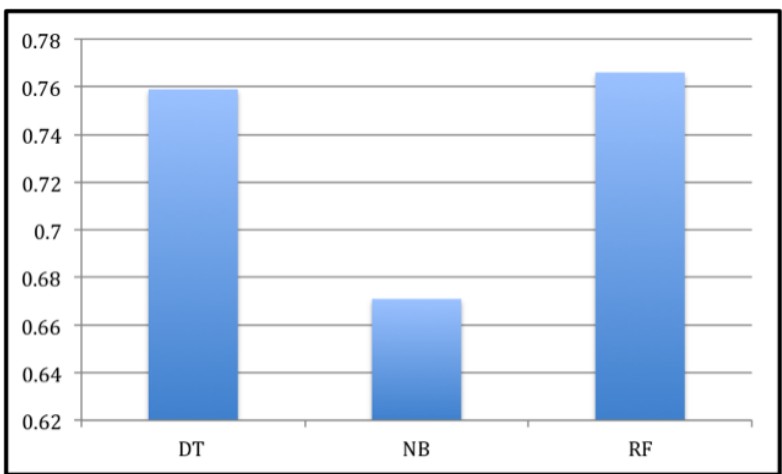

**Figure 16.** F-measure for DM techniques.

As shown in Figure 17, three scenarios from the proposed model outperformed traditional data mining techniques. Naïve Bayes has the lowest value of 0.67, while it increased, when ensemble methods were applied, to 0.72 (with boosting). Boosting with a decision tree achieved the highest value of recall at 0.78 compared to the traditional DM techniques, with a value of 0.77.

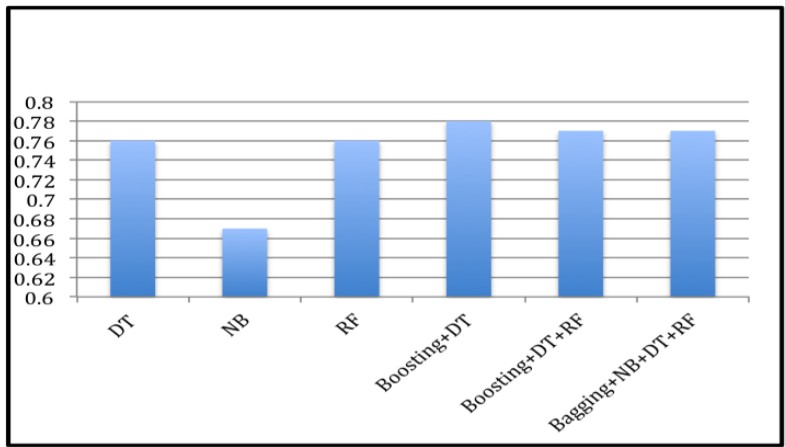

**Figure 17.** The F-measure of DM techniques and ensemble methods.

*5.5. Validate the Results with Previous Studies*

Validation is essential when creating predictive models because it identifies how reasonable they are. In this study, we compare the results with previous studies that used ensemble methods to predict student performance. The experimental value reports the betterment of proposed model over recent approaches as the accuracy was increased by 1%. The betterment is illustrated also when the voting technique was applied, the researchers increased the accuracy by 1% [19]. In [14], the same data set was used in a study that employed ensemble methods approaches. They increased the accuracy by 2.1% by using the boosting method with a decision tree. Likewise, with regard to [15], researchers aimed to improve student performance using ensemble methods. However, in [31], the results indicated that the ensemble methods improved accuracy by 1% when applied to the same data set. In a study that combined traditional data mining with the help of the voting method, the proposed model enhanced accuracy by 2.1% using the exact data set with ensemble methods [32]. Therefore, it is confirmed that the proposed model in this study has showcased improved performance over the existing approaches. Furthermore, our model enhanced performance by 2.4% when applied to a decision tree using the boosting

method. In the future, an ensemble fusion-based DL model can be developed to improve the performance of the proposed technique.

## 6. Conclusions

Academic institutions all over the world are concerned about student success. As learning management systems become more widespread, an enormous amount of information about the interaction between the teaching and learning processes is generated. In this study, the authors developed a new technique for predicting student performance that combined data mining techniques with ensemble methods. The data sets measured various features such as demographic information, student behavior in online classes, and parental involvement in academic performance. According to the findings of the study, there is a strong relationship between student behavior and their performance. For ensuring the enhanced performance of the proposed model, wide-ranging experiments were performed, and the results are inspected under distinct aspects. The proposed model improved the accuracy from 0.75 to 0.77 when we used the DT method with boosting, which resulted in a more accurate prediction of student performance. Furthermore, the precision and recall results both improved from 0.76 to 0.78. Moreover, the extensive experimentation outcomes confirmed the superior performance of the proposed technique compared to other existing techniques. Thus, the proposed technique can be used as a proficient approach for the prediction of student performance. In the context of future development, the presented model can be extended Utilization of ensemble fusion-based DL and data mining techniques for the prediction of the academic performance of students. Moreover, the proposed model can be extended to improve the performance of the proposed technique to several online educational data sets to support decision-makers for high-impact e-learning development.

**Author Contributions:** Conceptualization, A.A.A., A.S.A. and M.R.; methodology, A.A.A. and M.R.; software, A.A.A.; validation, A.A.A., A.S.A. and M.R.; formal analysis, A.A.A. and A.S.A.; investigation, A.A.A. and M.R.; resources, A.A.A., A.S.A. and M.R.; data curation, A.A.A. and A.S.A.; writing—original draft preparation, A.A.A.; writing—review and editing, A.A.A. and M.R.; visualization, A.A.A.; supervision, A.S.A. and M.R.; project administration, M.R.; and funding acquisition, A.A.A., A.S.A. and M.R. All authors have read and agreed to the published version of the manuscript.

**Funding:** This research work was funded by Institutional Fund Projects under grant no. (IFPIP: 26-611-1443).

**Institutional Review Board Statement:** Not applicable.

**Informed Consent Statement:** Not applicable.

**Data Availability Statement:** The data used in this work are available at Kaggle.

**Acknowledgments:** This research work was funded by Institutional Fund Projects under grant no. (IFPIP: 26-611-1443). Therefore, the authors gratefully acknowledge technical and financial support provided by the Ministry of Education and Deanship of Scientific Research (DSR), King Abdulaziz University (KAU), Jeddah, Saudi Arabia.

**Conflicts of Interest:** The authors declare no conflict of interest.

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
