# Peer review of "Enhancement of E-Learning Student’s Performance Based on Ensemble Techniques"

_electronics, doi:10.3390/electronics12061508_

Round 1

Reviewer 1 Report

1. Figure 7-9 need to be improved significantly, they are difficult to read in the current resolution.

2. There is no mention about ethics procedure of the study, whether the study needs ethics approval.

3. Section 5 needs to be improved significantly. The authors should discuss their results with previous studies and the current literature as they presented in the introduction. Currently, they only mentioned two studies in 5.5.

Author Response

Author’s Response to Reviewer 2 Comments

We would like to thank the reviewers and editors for providing an opportunity to revise the manuscript. We have studied these comments carefully and have made corresponding corrections that we hope will meet with your approval. In addition, as per the reviewer comment, we have improved the language quality of the manuscript and thoroughly proofread for grammatical as well as typographical errors. The revised text is mentioned in red font color.

Reviewer 2:

  • Figure 7-9 need to be improved significantly; they are difficult to read in the current resolution.

As per the reviewer comment, the figures quality is enhanced in the revised manuscript and became clearer and more legible by increasing their size. Kindly refer Figures 7, 8, and 9.

  • There is no mention about ethics procedure of the study, whether the study needs ethics approval.

Thank you for the comment. The study needs no ethical approval since it does not involve human participation in the experiments. The data set is commonly used and publicly available on Kaggle.

  • Section 5 needs to be improved significantly. The authors should discuss their results with previous studies and the current literature as they presented in the introduction. Currently, they only mentioned two studies in 5.5.
  • Based on the reviewer comment, we have provided more comparisons in the revised We have made a detailed comparison study of the proposed model with recent methods under different measures. Kindly refer section 5.5.

The authors combined traditional data mining with the help of voting method, the proposed model enhanced accuracy by 1% using the exact data set with ensemble methods [30]. Also, in [31], the results indicated that the ensemble methods improved accuracy by 2% when applied to the same data set. Furthermore, when the voting technique was applied, the researchers increased the accuracy by 2% [32].

  • As per the reviewer comment, recent references are provided in the revised manuscript. Kindly refer Page 20, References.

  • Improve the language quality.

As per the reviewer comment, we have improved the language quality of the manuscript and thoroughly proofread for grammatical as well as typographical errors.

We thank the reviewer for the positive comments.

Reviewer 2 Report

1. What is the main question addressed by the research?
The main question was to analyze and predict student performance using information gathered from online systems using three data mining models, namely: Decision Tree, Naive Bays, and Random Forest
2. Is it relevant and interesting?
The research is relevant to the existing literature, applies netnographic methods and provides novel treatment of data to deepen the understanding of how students learned through e-learning during the pandemic time
3. How original is the topic?
The article seems original in three ways: (1) using qualitative methods for data analysis, (2) implementing three data mining models to investigate the research problem & (3) referring to updated literature accounts in the area under investigation
4. What does it add to the subject area compared with other published material?
Using the three data mining models to inquire about the research problem supported with the qualitative analysis of data adds new ideas/paradigm to the current published work
5. Is the paper well written?
Yes, the syntax and typography seem excellent
6. Is the text clear and easy to read?
The text readability is legibly scientific, sound and clear
7. Are the conclusions consistent with the evidence and arguments presented?
Yes, as far as the reviewed literature allows. However, there should be more improvements that need to be made
8. Do they address the main question posed?
Yes, to a great extent
Improvements required are: 1. Add more literature from the study context (environment) 2. Discuss/Interpret the results in scope of these new studies and contextualize them

Author Response

We would like to thank the reviewers and editors for providing an opportunity to revise the manuscript. We have studied these comments carefully and have made corresponding corrections that we hope will meet with your approval. In addition, as per the reviewer comment, we have improved the language quality of the manuscript and thoroughly proofread for grammatical as well as typographical errors. The revised text is mentioned in red font color.

Reviewer 1:

  • Add more literature from the study context (environment)

Thank you for the suggestion. We have made a detailed related study of the proposed model with recent methods under different measures. As per the reviewer’s comment, we have added some recent related literature to enhance the related work in the revised manuscript. Kindly refer the page n. 4.

  • contextualize them.

Thank you for the comment. As per the reviewer’s comment, the validation of the results with previous studies has been updated and the reference to the compared methods are given. Kindly refer the page n. 18.

  • Improve the language quality.

As per the reviewer comment, we have improved the language quality of the manuscript and thoroughly proofread for grammatical as well as typographical errors.

We thank the reviewer for the positive comments. 
